# Coexistence rules for small, antagonistically interacting microbial communities

Gaurav S. Athreya[1¤a], Chaitanya S. Gokhale[2,3], Prateek Verma[4,5¤b]*

1 Indian Institute of Science Education and Research, Pune, India, 2 Research Group for Theoretical Models of Eco-evolutionary Dynamics, Department of Evolutionary Theory, Max-Planck Institute for Evolutionary Biology, Plön, Germany, 3 Chair for Computational and Theoretical Biology, Julius-Maximilians University Würzburg, Würzburg, Germany, 4 Institute for Chemistry and Biology of the Marine Environment (ICBM), Carl von Ossietzky Universität Oldenburg, Oldenburg, Germany, 5 Faculty of Business Administration and Economics, Bielefeld University, Bielefeld, Germany

¤a Current address: Institute of Organismic and Molecular Evolution (iomE), Johannes Gutenberg University, Mainz, Germany
¤b Current Address: Divisions of Epidemiology and Biostatistics, School of Public Health, University of California, Berkeley, California, United States of America
* prateekverma@berkeley.edu

## Abstract

The coexistence of diverse microbial communities despite the common presence of antimicrobial weapons presents a fundamental puzzle in ecology. To address this issue, we investigate the role of antibiotic-mediated interactions in driving microbial diversity using methods from graph theory and theoretical ecology. Our exhaustive analysis of small interaction graphs involving antibiotic production, resistance, and degradation reveals that two factors together determine whether an interaction pattern can support coexistence or not: a certain producer-sensitive-degrader (PSD) motif is critical, and a form of cyclicity between the action of different antibiotics is necessary. Using individual-based simulations, we also explore the role of the PSD motif in spatially structured populations and demonstrate that community coexistence is robust over a wide range of antibiotic and degrader diffusivities. Our findings provide a deeper understanding of the interaction patterns that drive diversity in complex microbial communities. Specifically, we emphasize how antagonism does not preclude biodiversity. These results offer clear pathways for cultivating synthetic microbial consortia, enabling the design of more effective strategies for manipulating microbial communities.

## Author summary

Antibiotics have revolutionised our society by their property of killing or attenuating bacteria. However, despite being widespread, antibiotics have a gentler influence in shaping microbial communities in the wild. We highlight the specific interactions between microbes essential for preserving a diverse community by elucidating

**Data availability statement:** The source code and data used to produce the results and analyses presented in this manuscript are available from the Zenodo repository: https://doi.org/10.5281/zenodo.17736401.

**Funding:** The author(s) received no specific funding for this work.

**Competing interests:** The authors have declared that no competing interests exist.

the minimal rules of engagement related to antibiotic sensitivity, production and degradation. Our results show that such bacterial rivalries can indeed coexist and that the presence of a specific kind of interaction is crucial for community stability. These findings provide a fresh perspective on influencing and steering microbial communities with implications for intervention strategies in the biotechnological development of microbial consortia.

## Introduction

It is universally acknowledged that microbial ecosystems can harbour immense biodiversity [1–4]. Explaining this coexistence of several species has long been a major goal of Ecology [5–7]. Historically, the competitive exclusion principle, based on the critical assumptions of resource scarcity and a static environment, has guided ecological research for many decades [8–11]. The competitive exclusion principle is built on the observation that two species that are identical in their ecological requirements cannot coexist indefinitely. This implies that the persistent coexistence of two species requires differences in their requirements, for instance, by specialising on [12] or being limited by different resources [13]. However, in the style of argument of Levine et al. [14], coexistence in extremely species-rich microbial communities, under this principle, would require a vast number of distinct "resource axes" on which species differ from each other. A central thread in ecological research, it has been argued that this resource competition-centric view of community coexistence, with roots in its conceptual exchanges with the (necessarily) anthropocentric field of economics—needs to have a broader view, most recently by Simha et al. [11].

Many bodies of work have arisen to offer a resolution to this "diversity paradox", such as those concerned with the importance of temporal heterogeneity in species performance [15], historical contingency and priority effects [16], metacommunity structure [17], and how evolutionary processes affect character displacement [18]. One promising subfield focuses on whether, and if so, which patterns of interactions between species can give rise to community stability. Even here, theoretical methods suggest that large interacting communities, intrinsically by their size, are likely unstable [19]. Further, there is empirical evidence from studies of interaction networks that competitive, and not cooperative, interactions might be more prevalent [20], and there are numerous mechanisms by which microbes inhibit each other's growth (reviewed in [21,22]). High microbial diversity thus appears to be counterintuitive.

Much of this work, however, focused on interactions being purely pairwise. Pairwise information is at least sometimes sufficient [23], but the more general explanation might lie in the existence of higher-order interactions [14]. This idea has received much attention: theoretical work has been developed [24,25], and indeed, it has been shown empirically that such interactions are not only prevalent but also important. Coexistence can emerge at the community level even though it is absent when species are present in pairs [26].

It is thus complicated to disentangle the joint consequences of antimicrobial weapons and higher-order interactions on population dynamics, particularly the coexistence outcomes they lead to. In this work, we focus on the consequences of a particular form of a higher-order interaction—the production, resistance, and degradation of chemical toxins that result in antagonistic interactions, here used interchangeably with "antibiotics" [21,27,28]. Via a mathematical model and computer simulations, we endeavour to understand which patterns of antibiotic-mediated interactions are necessary and sufficient for community coexistence.

An antibiotic, hereafter, is any compound secreted by a microbe that leads to a decrease in the growth rate of other microbes, either by killing them (bactericidal) or by stalling cellular processes that prevent further growth (bacteriostatic). Unilaterally referring to all molecules humans have used as antibiotics as "weapons" is reductive—some such secretions can have varied functions, such as signalling and growth stimulation when present in low concentrations and are, therefore, not always responsible for inhibition [29–31].

The strains in microbial communities can produce and be resistant or sensitive to several antibiotics [32,33]. The pathways that enable these properties are also modular, with genes often found on plasmids, making them prone to horizontal gene transfer [34,35]. Degradation of antibiotics is known to be a common mechanism of resistance [36,37], and has empirically been shown to have definite ecological and evolutionary consequences [38,39]. Numerous community configurations are possible since each strain can have a different phenotype concerning each antibiotic. Empirical investigations of the interaction graphs of soil-dwelling microbes have found many properties that are non-trivial and distinct from other ecological networks [40,41].

In this context, we are interested in finding general principles of interaction patterns that can sustain diverse communities. Coexistence in the face of antagonistic interactions has long been studied theoretically, such as in the work of Rescigno [42]. May and Leonard [43] showed that cyclical negative interactions between three species could lead to their dynamic coexistence in a well-mixed setting. This path to coexistence has some empirical support [44,45] but requires spatial structure [45–47], and is in principle unrealistic in a well-mixed setting since abundance is assumed to go arbitrarily close to zero—an unlikely situation in finite populations [48].

Kelsic et al. [49] showed that sustained coexistence is possible in the case of antibiotic-mediated interactions, even in well-mixed populations. In particular, the presence of strains that attenuate the effect of antibiotics, which is a type of higher-order interaction, can lead to refugia for sensitive strains that promote coexistence (Fig 1c). They studied in detail a three-strain community with cyclic antagonistic interactions, which showed robust coexistence (see Fig 1b) [49]. Sustained coexistence is, as expected, also observed in other communities with antibiotic-mediated interactions when there is spatial structure [45,50–53]. There is a feature common to most stably coexisting communities found in this body of work: their interaction graphs involve a producer-sensitive interaction between two species with a degrader strain attenuating this interaction. We refer to this structure as a producer-sensitive-degrader (PSD) motif. The importance of antibiotic degradation is also evidenced by the experiments of Abrudan et al. [54], who find widespread suppression of other strains' antibiotic production in a coexisting community. They also found widespread induction of antibiotic production, but the coexistence of many strains was never possible under only induction, except due to spatial structure. It is therefore imperative to understand the relevance and mechanistic underpinnings of the PSD motif in promoting coexistence.

We search all possible communities with a fixed complexity (strains × antibiotics) for stable steady-states and then vary this complexity. Thus, checking all possible interaction graphs allows us to infer exact conditions for stability. The observation of the PSD motif naturally lends itself to a fundamental question: given a community of $N$ strains that can interact via $M$ distinct antibiotics, what interaction patterns are necessary and/or sufficient for their coexistence? To answer this, we systematically study the population dynamics induced by various interaction graphs across different community sizes and complexity levels. Given the importance of space in structuring microbial population dynamics [55–58], we explore its effects by focusing on a community which does not exhibit a stable fixed point in a well-mixed scenario.

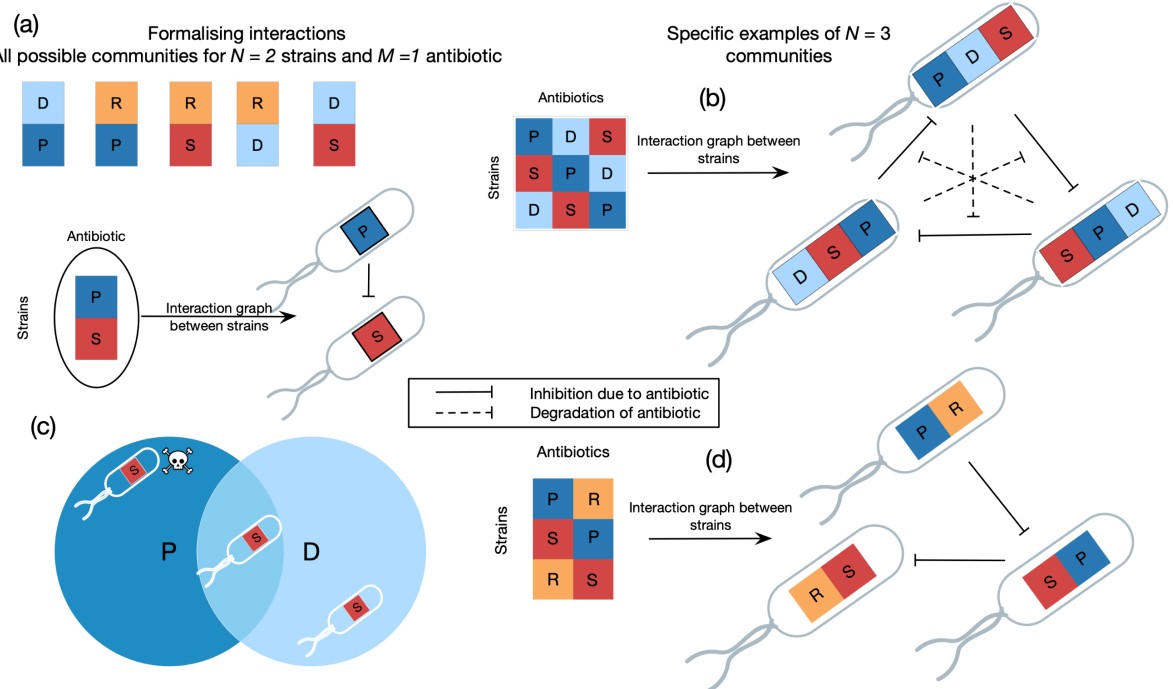

**Fig 1. Description and representation of model microbial communities.** We formalise interspecies interaction graphs with discrete phenotypes — producer (*P*), sensitive (*S*), degrader (*D*), or resistant (*R*) to an antibiotic. Panel (a) showcases the 6 unique communities with $N = 2$ distinct strains interacting via $M = 1$ antibiotic. The problem of how many distinct communities can exist for any $N$ and $M$ is treated in the Results section, and in the Sect A1 of S1 Appendix. (c) The effect of degradation as a mechanism of resistance. Any sensitive strains that are in the vicinity of a degrader are not affected by the antibiotic. In (b,d), we present the interaction graph of specific examples of communities. The skull icon in panel (c) is adapted from public-domain artwork on Openclipart (https://openclipart.org/detail/121987).

## Results

### Ecological dynamics: Mixed-inhibition zone model

We use the mixed-inhibition zone model of Kelsic et al. [49] with discrete phenotypes - a strain can be a producer (P), degrader (D), intrinsically resistant (R), or sensitive (S) to a particular antibiotic. We visualise a laboratory experiment with all strains present together in a flask, complete mixing of the individuals, and then dilution after a fixed period—we are modelling microbial abundances from one flask to the next, leading to discrete, non-overlapping "generations".

Antibiotic production, intrinsic resistance, and degradation are metabolically expensive, with the highest cost for production followed by degradation and then resistance. This cost structure is assumed because producers produce antibiotics and are resistant to them, degraders secrete chemicals that protect several individuals, whereas intrinsic resistance benefits only one individual. Assuming a constant base growth rate for all strains, producers thus have the smallest growth rate and sensitive strains the largest. Unless mentioned otherwise, we assume that the costs of exhibiting a particular phenotype (P,D,R) concerning a given antibiotic are similar across the antibiotics, but not exactly identical. We implement this by fixing a cost of resistance $c_r$, and adding a small random number to it for each antibiotic. The scale of this random number is chosen so that it removes any edge-case results that arise only because of identical costs, while still retaining the dynamical behaviour of interest. In particular, we choose the realised cost of resistance for each antibiotic as $c_r + 0.0005 \times \text{Uniform}[-1, 1]$. The relative costs of antibiotic degradation and production, $c_p/c_r$ and $c_d/c_r$ are then varied. When comparing communities, we set the costs back to identical to only focus on the effect of the interaction graphs.

To model the effect of the antibiotic and degrader chemicals, we assume, following [49], that there is a critical region around an individual where the chemical is effective. We relax this assumption in later sections and consider an explicitly diffusive process. Here, we model the population dynamics using a discrete version of the replicator equation [59]

$$X_i(t+1) = \frac{f_i(t)X_i(t)}{\sum_{i=1}^{n} f_i(t)X_i(t)} \tag{1}$$

The fitness $f_i$ of a strain at time step $t$ is written as the product of its growth rate with the probability that it is not killed due to antibiotic sensitivity

$$f_i(t) = r_i(1 - p_i^{kill}(t)) \tag{2}$$

as based on the phenotype of the strains [60]. See the Methods section for the computation of $p_i^{kill}(t)$.

Using the developed model, we analyse communities with different numbers of strains and antibiotics. For a given $N$ and $M$, the set of all unique communities with $N$ strains and $M$ antibiotics can be enumerated by the algorithm outlined in Sect A1 of the S1 Appendix. We explore this set of graphs and determine whether (and if so, which) communities with that many strains and antibiotics can exhibit stable states. Our objective is to search for communities with stable internal fixed points; such points correspond to a steady-state where all strains coexist. We find these fixed points by numerically solving a system of equations for each community, starting from 100 randomly chosen initial conditions. The initial conditions are chosen from a Dirichlet(1) distribution on the ($N$–1)-simplex. The stability of a fixed point is determined by computing the associated Jacobian matrix—a fixed point is stable if the spectral radius of the Jacobian evaluated there is less than 1 (recall we are in a discrete-time setting).

In general, populations can also coexist when there are no stable fixed points via, for example, limit cycle oscillations; more generally, see the ideas of persistence and permanence in population dynamics [61,62]. In our work, we focus only on stable fixed points for their analytical and computational tractability. Previous work has shown that the likelihood of a dynamic steady-state decreases if the community members are close to each other in traits as opposed to an assembly of dissimilar members [63].

## Enumeration and analysis of interaction graphs under the mixed-inhibition zone model

It is possible to calculate the number of interaction graphs, equivalently the number of communities, with $N$ strains, $M$ antibiotics, and phenotypes taking on one of the 4 possible values (P,S,D,R). This number is given by (see Sect A1 in S1 Appendix, [64,65])

$$\frac{1}{N!M!} \sum_{i=0}^{N} \sum_{j=0}^{M} s(N,i)s(M,j)4^{ij} \tag{3}$$

where the $s(x,y)$ denote the Stirling numbers of the first kind. While this provides intuition, there does not exist (to the best of our knowledge) an efficient algorithm to actually construct all these distinct interaction graphs. Enumeration becomes an especially non-trivial task for higher complexity communities. In Sect A1 of S1 Appendix, we describe an algorithm for this task, and use it to exhaustively search through all possible interaction topologies for a given $N$ and $M$.

Since the space of communities becomes large as $N,M$ increase, we perform these exhaustive searches across network topology only at two parameter combinations. These combinations differ in their cost structures, i.e. how much more the costs of degradation and production are than the cost of intrinsic resistance. We use the values $K_P = 40, K_D = 10$ for the antibiotic and degrader strengths; these are assumed to be identical across the antibiotics. Cost of resistance is taken to be $c_r = 0.16$; one parameter combination is "low relative cost", i.e. $c_p = 2.5c_r, c_d = 1.4c_r$, the second combination is "high

relative cost", $c_p = 2.7625\ c_r, c_d = 2.2083\ c_r$. This affects the growth rates of the different strains in each community. The parameters are picked via, and the exhaustive-search results are supplemented by, thorough parameter sweeps on specific communities of interest (see Fig 2). Results are discussed below, and summarised in Table 1.

**Notation.** We denote communities with $N$ strains and $M$ antibiotics by an array of $N$ rows, with each row denoting a strain in the community. Each row is a string of $M$ letters from P, S, D and R, and the $j$th letter signifies the phenotype of the given strain with respect to the $j$th antibiotic. For example, the community on the right in Fig 1 is denoted by [PDS, SPD, DSP].

**Interacting via one antibiotic.** When strains interact via only one antibiotic, no count of strains is sufficient for the existence of a stable steady-state. In particular, the communities [P,S,D] (a community where one strain produces an antibiotic, another is sensitive to it, and a third degrades it; see methods for a full explanation of this notation) and [P,S,R] do not have any stable fixed points with all strains coexisting. All trajectories lead to one strain fixing in the

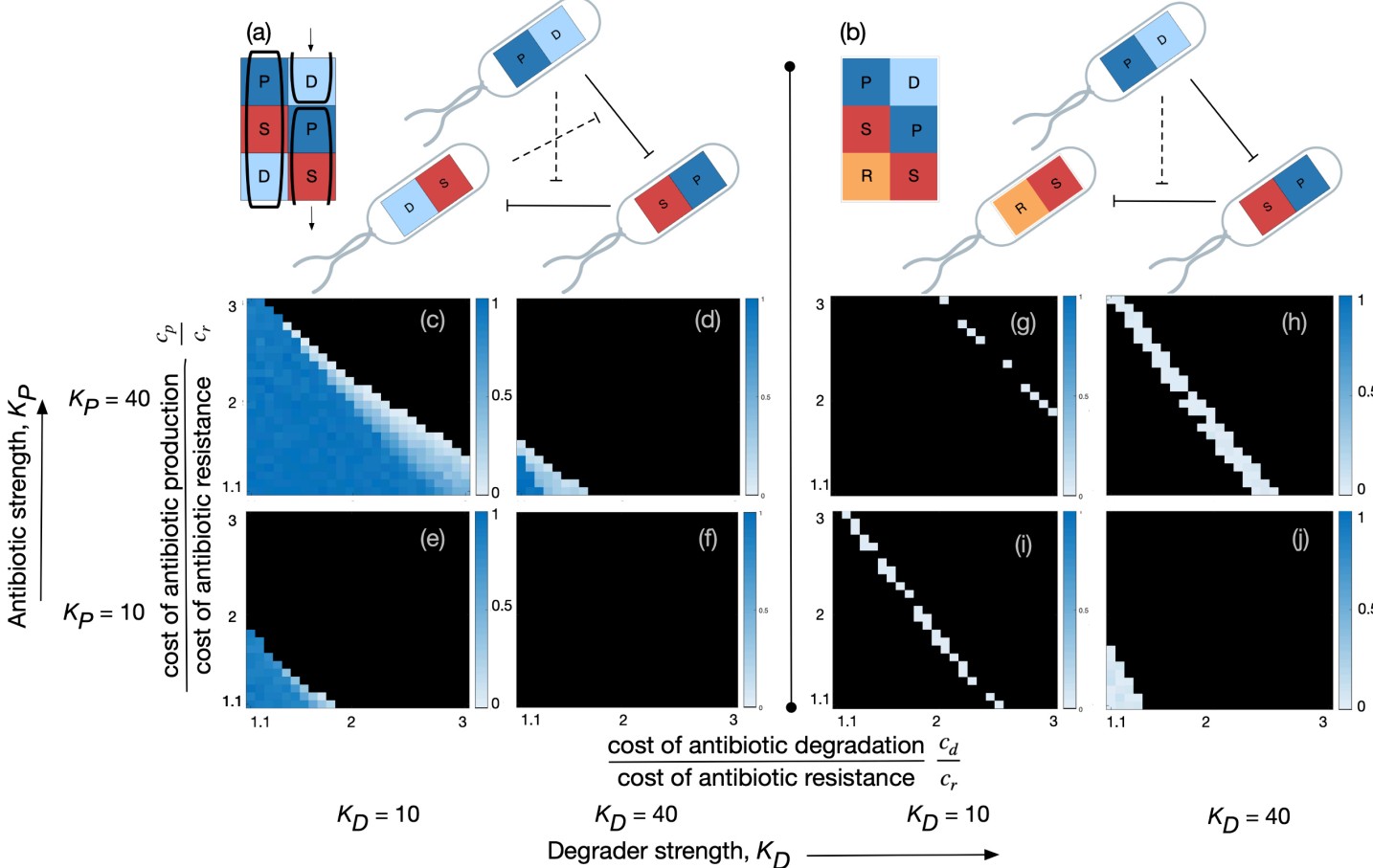

**Fig 2. Comparing the size of basins of attraction between the two communities that display coexistence with 2 antibiotics.** Three strains is the minimal complexity necessary for stable coexistence under the mixed-inhibition zone model, i.e. without spatial structure. The communities [PD,SP,DS] (a) and [PD,SP,RS] (b) are the only communities that we observe to coexist. Starting from 100 randomly chosen initial frequencies, we numerically iterate Eq (5) for community (a) across a wide range of metabolic costs, implemented by fixing $c_r$, and varying $c_p/c_r$ and $c_d/c_r$. Heatmaps (c-f) and (g-j) depict the fraction of trajectories that reach a stable, internal fixed point. Black indicates a fraction of zero. This procedure is carried out for a high and low value, each, of the antibiotic strength $K_P$ and degrader strength $K_D$. Metabolic costs and strengths are here assumed to be identical across antibiotics. Coexistence regions never overlap. Coexistence is most likely for community (a) when $K_P$ is high and $K_D$ is low; most likely for community (b) when both $K_P, K_D$ are high. *Parameters*: $g = 1.0, c_r = 0.16$, others as indicated in the figure.

**Table 1**. **For each *N,M* pair above, we explore the space of all possible communities of *N* strains interacting via *M* antibiotics.** We numerically solve the associated equations (see Methods, "mixed-inhibition zone model") describing a well-mixed population, and report *N,M* pairs for which there exists at least one interaction pattern that displays a stable, internal fixed point. This analysis is performed at two parameter combinations. Common parameters: $g = 1.0$, $c_r = 0.16$, $K_P = 40$, $K_D = 10$. Different between the two parameter combinations: $c_d = 1.4c_r$, $c_p = 2.5c_r$, or $c_d = 2.2083c_r$, $c_p = 2.7624c_r$. Parameter-dependence of the fixed points is discussed in the main text.

| No. antibiotics, *M* | No. strains, *N* | Stable, internal fixed point present? |
|---|---|---|
| 1 | 2 | No |
|  | 3 | No |
|  | 4 | No |
| 2 | 2 | No |
|  | 3 | Yes, 2, parameter-dependent (see Fig 2) |
|  | 4 | No |
| 3 | 3 | Yes, many, parameter-dependent (see Fig 3) |

population—in P, S, R it can be the resistant strain or the sensitive one, depending on parameters (Fig A5, a.1-3 in S1 Appendix), in [P,S,D] it is always the sensitive strain (Fig A5, b.1-3 in S1 Appendix). The results of [53] and [45] show that spatial structure induces coexistence in these communities.

**Interacting via two antibiotics.** The picture is more complex when there are two antibiotics. No combination of two strains can coexist; the presence of three strains is the minimal complexity required for a steady-state. Precisely two community topologies can have an isolated stable internal fixed point for our parameter combinations—the [PD,SP,DS] community, and [PD,SP,RS], both shown in Fig 2. However, each community supports coexistence only at one of the above parameter combinations—when metabolic costs are low, [PD,SP,DS] (Fig 2 left) is uniquely stable; when costs are relatively higher, [PD,SP,RS] (Fig 2 right) is stable. We investigated this result in more detail, performing parameter sweeps along $c_p/c_r$, $c_d/c_r$, $K_P$, and $K_D$ (Fig 2) and the regions of coexistence also do not overlap across this larger range. When the respective fixed points exist, the basin of attraction of [PD,SP,DS] is consistently much larger. Trajectories of population dynamics are presented in Fig A7(a,e) in S1 Appendix.

Both these communities involve a producer, a sensitive strain, and a degrader/resistant strain. The community [PD,SP,DS] has a producer, degrader, and sensitive strain corresponding to both antibiotics - a PSD motif. Moreover, the pattern of these interactions is also "cyclic", i.e. the matrix describing this community is a circulant matrix: (i) the columns consist of the same elements, and (ii) each column is rotated one element downward compared to the previous one (see Fig 2a for a depiction). Given the three phenotypes P, S, D, and a community interacting via 2 antibiotics, the above community is the only community that can have this circulant form. We shall refer to it as the circulant community with 2 PSD motifs, and the property of the matrix being circulant as cyclicity in the interactions. The magnitude of cyclicity of an interaction matrix is the maximum number of columns that have this circulant pattern.

The community [PD,SP,RS] displays stable coexistence at relatively higher metabolic costs. It does not have a PSD motif for both antibiotics; the more "private" form of resistance *R* is here sufficient to support coexistence. However, the interaction pattern has a circulant structure reminiscent of the above—in fact, replacing the "D" in the third row of [PD,SP,DS] with an "R" yields exactly the circulant community above.

Two communities are now natural to investigate: the community obtained by replacement of the other D with an R ([PR,SP,DS]), and the circulant community containing two PSR motifs ([PR,SP,RS], interaction graph in Fig 1c). Interestingly, neither community supports coexistence, even when searched on the larger parameter range in Fig 2. In the graph that supports coexistence, the strain producing an antibiotic also degrades the antibiotic produced.

This analysis shows that at least one PSD motif is necessary, and that even when the second antibiotic has a PSR motif, the two positions that the R can take are not equally important in inducing coexistence. Cyclicity of interactions both within and between across the action of different antibiotics, however, seems to be necessary.

We also explored the space of communities with $N = 4$ strains. No communities displayed coexistence at either parameter combination.

**Interacting via three antibiotics.** Here we only explore communities having up to $N = 3$ strains; this is also the minimal number of strains required for coexistence when $M = 3$. There are many communities with a stable internal fixed point with $N = 3, M = 3$: at the low cost parameter combination, 14 out of the 6864 total communities display stability; at high costs, only 3 interaction graphs support coexistence. The latter three communities are a subset of the former 14.

To describe these communities, we first define the notion of an "extension" of a community: given a community C, an extension of C is another community whose interaction graph can be expressed as that of C, but including interactions via another antibiotic. Stably coexisting communities fall into four categories, three of which are easy to comprehend: the circulant community with 3 PSD motifs (1 of 14, Fig 3c), extensions of the circulant community with 2 PSD motifs (10 of 14, example in Fig 3b), and extensions of the near-circulant community [PD,SP,RS] (2 of 14, example in Fig 3a). Both classes of extensions that support coexistence consist only of P, D, or R phenotypes along the new antibiotic—they do not introduce new antagonistic interactions (since all three of these phenotypes are resistant); instead, they act effectively as reductions in the growth rates due to the metabolic cost of producing/degrading/being resistant to another antibiotic. We expect the list of stable extensions to change according to the metabolic costs, but the presence of extensions as stable $N = M = 3$ communities is likely more generic. At high costs, the 3 communities that still survive are the circulant community with 3 PSD motifs, and two extensions of the circulant community with 2 PSD motifs.

In addition to the three categories above, we find stable coexistence in one community with 3 PSD motifs not arranged in a cyclic pattern (this can also be expressed as an extension of the circulant 2-PSD community). This community is

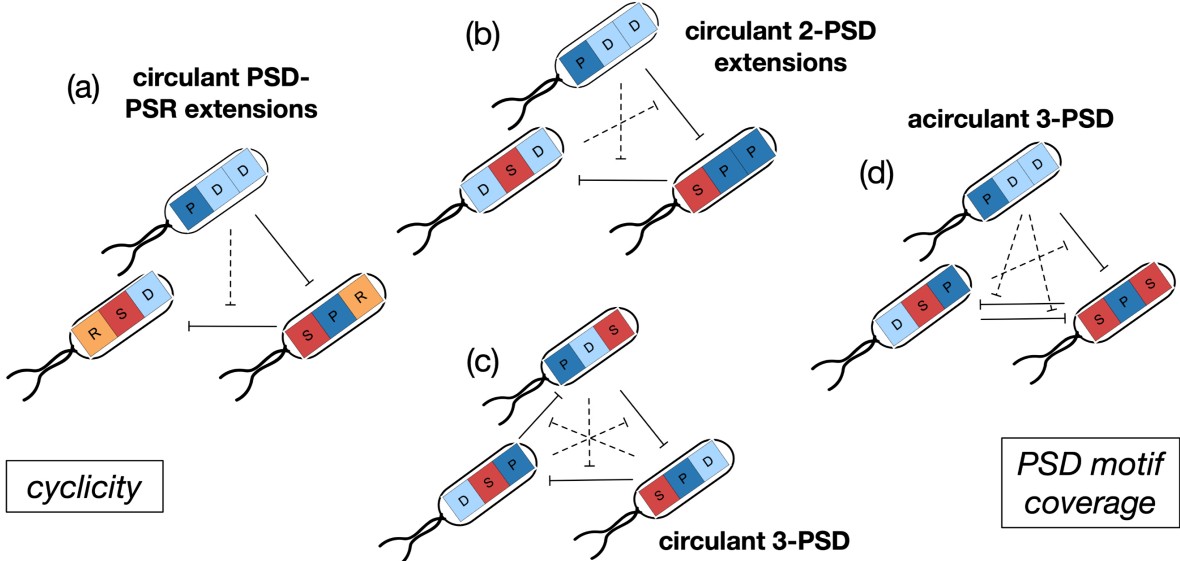

**Fig 3. A representation of the stably coexisting communities with N = 3, M = 3 under the mixed-inhibition zone model.** As described in the main text, this set of communities can be separated into 4 categories: (a) extensions of the circulant community with one PSD and one PSR motif, (b) extensions of the circulant community with two PSD motifs, (c) the circulant community with three PSD motifs, and (d) one acirculant community with 3 PSD motifs. Examples of these categories are represented in terms of the cyclicity in the interactions, and along how many of the antibiotics are present PSD motifs ("PSD motif coverage"). All communities are circulant across at least 2 of 3 antibiotics, and all communities have a PSD motif along at least 1 out of 3 antibiotics. They are depicted in this figure to reflect this: on the left is a coexisting community with only one PSD motif but cyclicity; on the right is a community with three PSD motifs but cyclicity along only 2 of 3 of these communities. The search was performed at two parameter combinations, we focus here on the one where more communities displayed coexistence, with parameters $g = 1$, $c_r = 0.16$, $c_d = 1.4\, c_r$, $c_p = 2.5\, c_r$, $K_P = 40$, $K_D = 10$. Metabolic costs are similar across antibiotics, but not identical. See section on mixed-inhibition zone model in the main text for details.

presented in Fig 3d, and importantly, it is still circulant along two of its antibiotics. Trajectories of population dynamics for an example from each of these categories are presented in Fig A7(b,c,d,f) of S1 Appendix.

These results show that a combination of cyclicity and PSD motifs is necessary for coexistence - every community displaying coexistence contains circularity along at least 2 antibiotics, and every community displaying coexistence contains at least one PSD motif (and including more PSD motifs results in a much bigger basin of attraction, Fig 2).

**Effects of interaction complexity and cyclicity.** We perform two parameter sweeps to further place these results in context. First, we will compare the circulant communities with 2 and 3 PSD motifs to evaluate the effect of interaction complexity (they differ only in their number of antibiotics, $M = 2, 3$ respectively). Second, restricting ourselves to $M = 3$ communities, we compare circulant and acirculant communities with 3 PSD motifs to evaluate the effect of cyclicity in interactions. We plot two measures of stability: the size of the basin of attraction (Fig 4, bottom row of panels), and the robustness to dynamical perturbations, i.e. 1 minus the spectral radius of the appropriate Jacobian matrix evaluated at the fixed point (Fig 4, middle row of panels). A higher robustness implies a quicker return to stable coexistence upon a slight change in abundances that pushes the population away from the steady state. We set robustness to zero if a fixed point does not exist or is unstable.

Comparing columns (a) and (b) shows the effect of complexity: the circulant community with 3 PSD motifs has a stable, internal fixed point for all values of $c_p/c_r$ and $c_d/c_r$, which is not true for the circulant communities with 2 PSD motifs. When the fixed points exist, the 3-PSD circulant community also has either a similarly sized or larger basin of attraction than the 2-PSD circulant community, irrespective of the metabolic costs. However, the circulant community with 3 PSD motifs is less robust to perturbations.

Comparing columns (b) and (c) shows the effect of cyclicity: again, the circulant community with 3 PSD motifs has a stable, internal fixed point across more parameter values than the acyclic community with 3 PSD motifs. When the fixed points exist, both fixed points have similarly-sized basins of attraction. However, we see the same pattern in the robustness: when there exists a stable fixed point, the acyclic community often has a higher robustness to dynamical perturbations. The circulant community with 3 PSD motifs has a robustness that is independent of $c_d/c_r$ and $c_p/c_r$, so we also calculated the robustness across $K_P$ and $K_D$. The robustness is highest for high $K_P$ and intermediate $K_D$, as reported by Kelsic et al. [49], however, it never goes higher than the values reached by the acirculant community in Fig 4c.

These comparisons suggest that the cyclicity in interactions provides advantages in terms of more generic existence of a stable fixed point and a large basin of attraction, but this simultaneously leads to slow establishment of, or return to, the steady state.

## Ecological dynamics in spatially structured populations

So far, the chemicals' impact has been considered by modelling circular inhibition zones of fixed radii within which they are fully effective and useless outside. A more natural treatment should account for the diffusion of degrader enzymes and antibiotics, with degraders being able to neutralise the effect of antibiotics closer to their location and less capable of doing so farther from their location. This concept has been explored in the context of the evolution of such systems, but the ecological conditions for coexistence have yet to be given any attention [53]. Further, previous literature provides strong evidence that the complicated conditions required for stability may be limited to well-mixed models [45,46].

As a first step in characterising this presumably larger-than-well-mixed class of stable communities, we study in detail one of its simplest members - the PSD motif with one antibiotic. The motif does not exhibit a steady-state under the well-mixed conditions in the mixed-inhibition zone model (Fig A5 of S1 Appendix). Further, as an instance of the PSD motif, this community's behaviour might lead to insights into how stability rules valid in the well-mixed case transform under the addition of spatial structure. For details of the implementation of the model, see Materials and Methods. Note that our model is not limited to any specific properties of the PSD motif, and the simulation method can be applied to any arbitrary community.

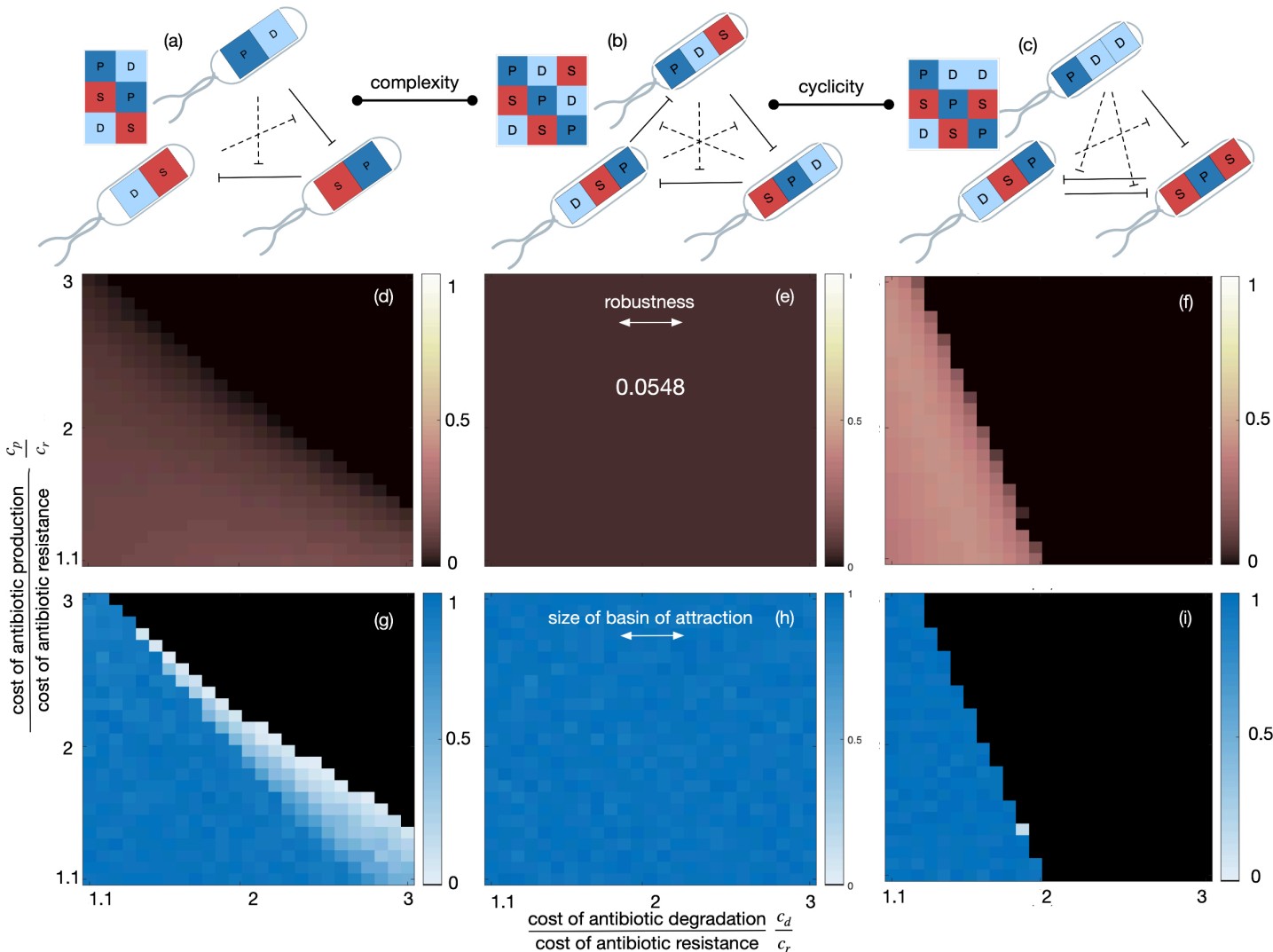

**Fig 4. The effects of interaction complexity and cyclicity.** We compare the stability of three communities to understand the effect of complexity and cyclicity. These comparisons are made in terms of two measures: (i) robustness to dynamical perturbations (d,e,f): for each parameter combination, we calculate the metric (1 minus the spectral radius of the Jacobian), evaluated at the fixed point. If a fixed point does not exist or is unstable, the robustness is set to zero; (ii) the size of the basin of attraction (g,h,i): for each parameter combination, we follow trajectories started from 100 randomly chosen initial conditions. The size of the basin of attraction is defined as the number of trajectories that settle at the internal fixed point. Community (a) vs (b) gives insights on the effect of interaction complexity; community (b) vs (c) on that of interaction cyclicity. *Parameters*: $g = 1, K_P = 40, K_D = 10, c_r = 0.16$, others as indicated in the figure. Metabolic costs are assumed equal across antibiotics.

## Results and analysis of the stochastic spatial model

**Spatial structure strengthens the PSD motif.** To study the effect of spatial structure, we perform individual-based simulations on a two-dimensional, hexagonal lattice. We include birth, death and dispersal of the microbes, and explicit diffusion of the antibiotic and degrader molecules that they produce. We study the ecological stability of this [P,S,D] community as a function of the diffusivities of the antibiotic and degrader chemicals, $K_P$ and $K_D$ respectively. While the $K_P$ and $K_D$ are meant to be similar in effect to the parameters introduced as part of the mixed-inhibition zone model above, there are some differences. The unidimensional "strength" of the chemicals in the mixed-inhibition zone model is split into

two notions of strength in the spatial model: the spread of the secreted chemical and the shape of the dose-response curve. Specifically, a high value of $K_P$ corresponds to a large diffusivity and a higher susceptibility to the antibiotic (see Sect A2 in S1 Appendix for details). The amount of the antibiotic produced (which is kept constant across individuals) and its diffusivity jointly determine the extent to which surrounding individuals experience it. The parameters $K_P, K_D$ do not change the metabolic costs, and only depend on external environmental factors.

Some examples of parameter values for which the population dynamics relaxed onto a stable state are shown in Fig 5. Fig 6 shows the result of these simulations over the explored $K_P - K_D$ parameter space in full. The abundances can be

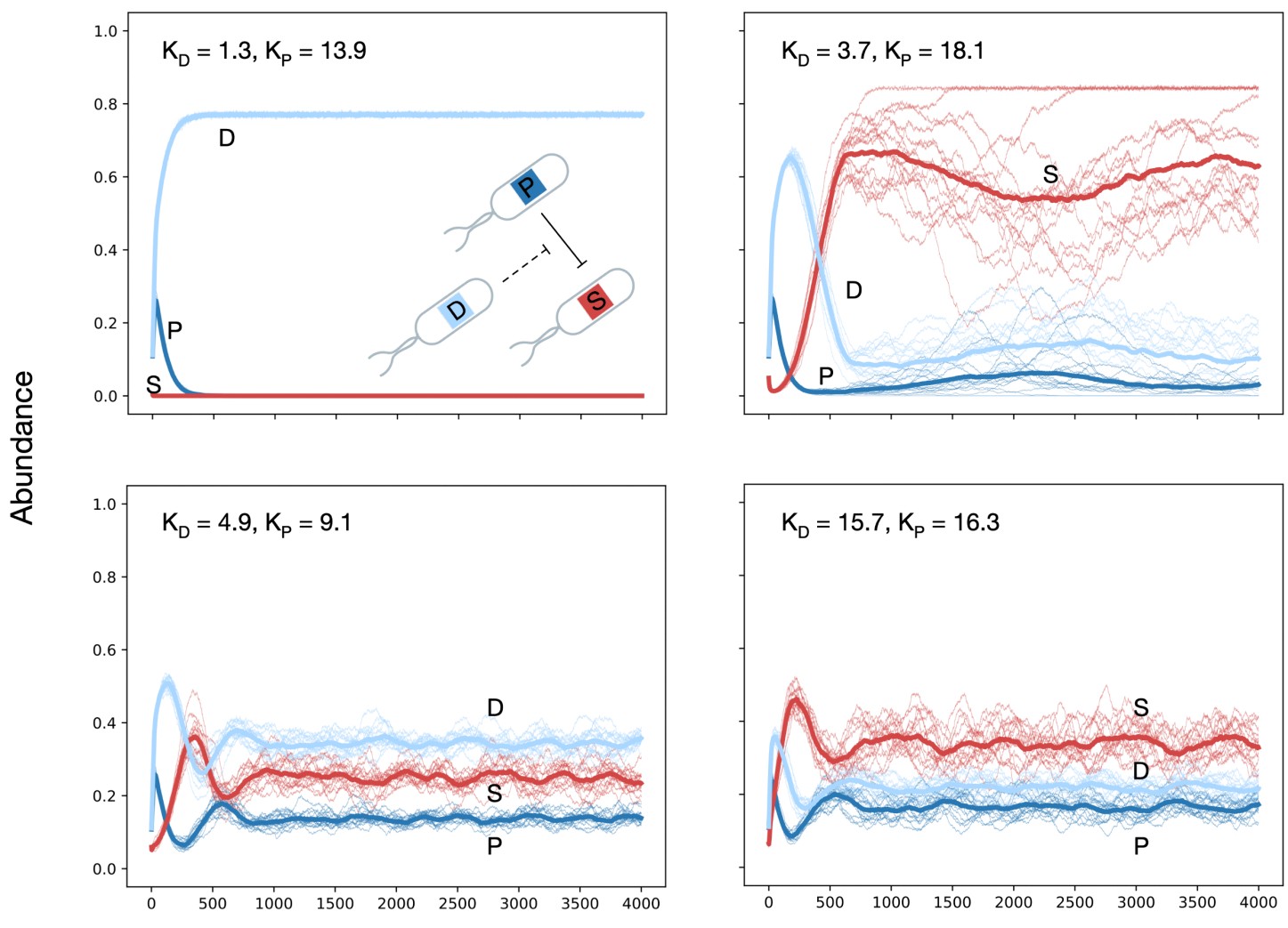

**Fig 5. Dynamics of a spatially structured population consisting of a PSD motif.** We conducted individual-based lattice simulations using periodic boundary conditions with the strains producer, sensitive, and degrader. In the four panels of this figure, we present examples of how the Producer, Sensitive, and Degrader strains change in abundance over time as they interact on the lattice. The four panels vary in $K_P$ and $K_D$ as indicated in the figure. Thin lines are independent simulation runs starting from a different initial configuration of individuals on the lattice; thick lines are averages across runs. Mean number of individuals of each strain at the beginning of the simulation is kept constant across these runs. Fig 6 illustrates the results of such simulations across a broad $K_P$ and $K_D$ spectrum. Parameter values: $g = 0.7, d = 0.3, c_r = 0.05, c_d = 0.105, c_p = 0.15$, grid size = 200, $u_p = 10, u_d = 10$.

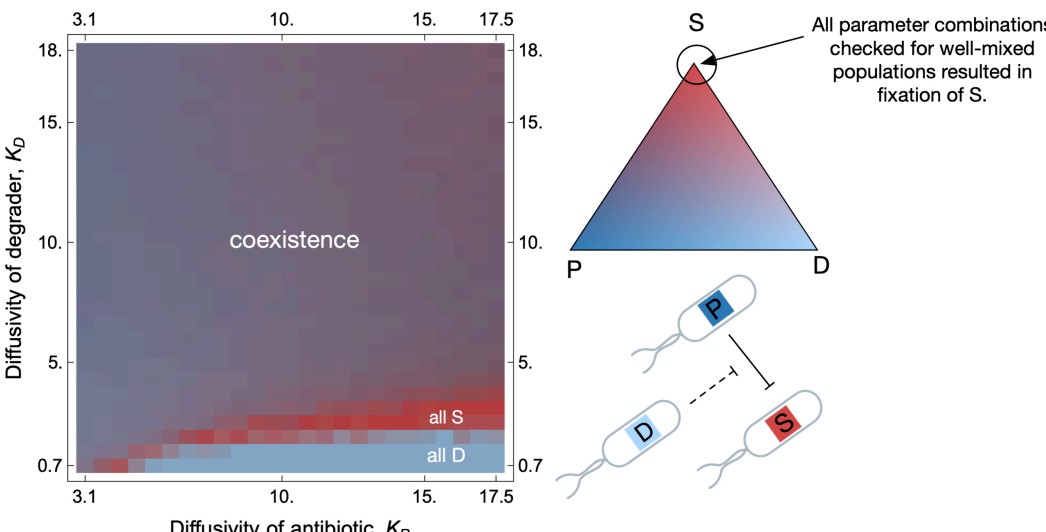

**Fig 6**. **Spatial structure fosters coexistence in the 1-antibiotic PSD motif**. We explore the population dynamics across antibiotic and degrader diffusivities, by performing 15 independent simulations across varying $K_P$ and $K_D$ values, over 6,000 generations. The initial lattice configuration varied across runs, maintaining only the fraction of the lattice occupied constant. The heatmap on the left summarises results: colour at a given $K_P$, $K_D$ coordinate denotes the average abundance of producer, sensitive, and degrader strains averaged over the last 500 time steps of a simulation. The colour must be read using the adjacent simplex, which acts as a legend, since it contains all numbers $(P,S,D)$ such that $P + S + D = 1$. Simplex vertices represent the maximal abundance of one strain: S (red), P (dark blue), and D (light blue), with mixed colours indicating coexistence. Unlike well-mixed PSD populations where the sensitive strain dominates, spatially structured populations exhibit non-zero strain abundances over a broad parameter range. Other parameters are the same as in Fig 5.

identified using the triangle on the right as a legend. We find a large coexistence region, indicated by mixed shades of blue and red, where all three strains persist. This stands in sharp contrast to well-mixed PSD populations, where the sensitive strain always grew to fixation (Fig A5 of S1 Appendix).

For large $K_P$ and small $K_D$, the sensitive strain is killed very quickly due to the high diffusivity of the antibiotic. Then, the producer is outcompeted by the degrader strain because of its lower metabolic cost. For slightly higher $K_D$, the degrader strain can protect the sensitive strain enough to prevent its extinction. Simultaneously, the degrader outcompetes the producer, after which the sensitive strain outcompetes the degrader. The deciding characteristic is the *timescale of local interaction* between patches of different phenotypes.

There is a clear boundary along which coexistence is lost to fixation by the sensitive strain (red in the heatmap, Fig 6 left), below which the degrader strain fixes (light blue in the heatmap, Fig 6 left). Along this boundary, one observes that the producer strain is always outcompeted first, and the eventual fate depends on whether or not a patch of sensitive individuals has persisted long enough for the producers to have died out (see movies, associated online repository). If sensitive individuals still exist, they outcompete the remaining degraders based purely on the difference in growth rates. If the producers have killed the sensitive strains, the degraders dominate due to their superior growth rate, as is observed in some simulation runs. The persistence (or lack thereof in some simulations) of the sensitive strain then determines the long-term fate of the population.

Based on the above explanation, we expect the location of this boundary to move with changes in the metabolic cost, but for the shape to stay similar. This suggests that a single steady state exists in this system's interior, which could be recovered in a deterministic model that tracks strain and chemical densities. Our extensions in this direction are promising but beyond the scope of the current manuscript.

The above-detailed analysis shows the complex interplay between growth and antibiotic concentrations that orchestrates the coexistence of these communities. This causal disentanglement is possible due to the simple nature of the community we are analysing. We have thus shown that spatial structure does not immediately imply coexistence. Still, the ecological stability of this specific community is nevertheless generic, especially when contrasted with the lack of stability in the mixed-inhibition zone model.

## Summary of results

Antibiotic-mediated interactions are widespread in nature and play key ecological roles [30,66]. When antagonistic chemicals are produced, they suppress certain strains, raising questions about how such communities maintain stability. We investigate the production and degradation patterns of antibiotics required for stable coexistence.

The interactions between the strains in a community can be formalised in terms of mathematical objects called graphs. We describe a strategy, limited only by computational power, to exhaustively analyse the dynamics induced by a wide range of interaction graphs. The ecological dynamics specified by each pattern are then analysed to determine the existence of a stable steady state for relative strain abundances. This procedure allows one to make a complete record of the behaviour of all interaction graphs. With these tools, we ask a simple question: which interaction graphs give rise to a stable fixed point for the community's population dynamics?

We show that, for a small number of antibiotics, simple principles can explain the existence of a stable state for community dynamics. From our analysis of communities interacting via one, two, and three antibiotics, we observe that a combination of two factors determines whether an interaction graph can support coexistence: these factors are (i) cyclicity in interactions, (ii) presence of a PSD motif. All communities had a circulant interaction matrix along at least 2 of the antibiotics, and all communities had at least 1 of 2, or 1 of 3 PSD motifs (Fig 2 and Fig 3). Communities with only one (of two possible) PSD motifs seem to have a much smaller basin of attraction (Fig 2). Interaction complexity and cyclicity favour coexistence over a wider range of parameters, however, this seems (Fig 4) to come at the cost of robustness to dynamical perturbations.

Microbes in natural communities likely interact via multiple antibiotics. When there is only one antibiotic to concern ourselves with, the dynamics is described by rock-paper-scissors dynamics, and is well-understood, e.g. see [45]. However, when there is an arbitrary number of channels via which microbes can interact, how can we explain the fact that they nevertheless coexist? We show that the interactions between strains must also follow a specific pattern: the effect of one channel (antibiotic) must be balanced by the effect of another. We formalise this by introducing the notion of a circulant interaction graph, and use this to classify interaction graphs by the magnitude of their cyclicity (Fig 3). We conjecture that the criterion of cyclicity between interaction channels being necessary for coexistence is general, and akin to the non-transitivity of interactions (P vs. S vs. R/D) that we observe is partially necessary within each antibiotic.

Lastly, we observed many coexisting communities that could be expressed as extensions of simpler (lower number of antibiotics $M$) communities. In most of these communities, the interactions along the new antibiotic involved only P,D, or R phenotypes. These phenotypes do not change the interaction graph, except that they impose new metabolic costs on the existing strains. It is, however, unlikely that these resistance phenotypes arise without there having first been a sensitive phenotype. This producer-sensitive interaction would be destabilising, and one must perform evolutionary simulations to speculate on whether or not the community reaches the P,D,R-extension that we observe as being stably coexisting.

To summarise, the outcome of this work is twofold: Firstly, it provides intuition for the kind of toxin-mediated interactions a stable community can be expected to have. This has clear implications for understanding the basic biology of microbial communities and for building intuition when synthetic communities are to be cultured in a laboratory. Secondly, we describe the design and implementation of a method to obtain exact conditions for coexistence despite a complex

dynamical system that is hard to analyse analytically. Improvements and modifications of this algorithm can potentially have a significant effect when considering larger classes of interaction structures. Such methods will only become more important in the future as successive stages of biological reality are introduced into models, making them progressively more intricate and applicable.

## Discussion

In this work, we characterised the mapping between the topology of the interaction network and coexistence outcomes. Explicitly including the strength of the antibiotics and degraders in our model and its discrete-time nature makes it challenging to obtain analytical results. This problem is exacerbated by our goal to establish results for ecological dynamics in a community with an arbitrary number of antibiotics and species and arbitrary interactions between them. An interesting way to circumvent this is to assume, following the work of May [19] and then, e.g. Allesina and Tang [67], that communities are large and interactions are randomly drawn from a distribution. This approach allows the application of results from random matrix theory that might yield deeper insights into the relevance of antibiotic production-degradation interactions in nature. This requires the development of theoretical methods as well, since higher-order interactions are described more appropriately not by a matrix, but by a tensor [68]. Czaran et al. [51], Bairey et al. [25], and Swain et al. [69] have begun this work by simulating such large systems. It would be an exciting extension of our work to connect such approaches and generalise these results analytically.

Adopting the mixed inhibition zone model of Kelsic et al. [49] has two consequences. Importantly, it does not explicitly model the concentrations of antibiotic and degrader molecules; this choice was made to (i) find a middle ground between being precise and computationally tractable and (ii) to ground our work in the already established work of Kelsic et al. [49]. However, Momeni et al. [70] have shown that phenomenological Lotka-Volterra models cannot capture the entire structure described by more precise mechanistic models. It would hence be ideal to use, e.g. consumer-resource models to describe the full effect of all strains on the concentrations of the antibiotics and degrader models (e.g. see [71–73]). The second consequence is that it produces artifacts in some simple cases because of the focus on only relative (and not absolute) abundances—if two strains produce antibiotics against each other, they might drive each other to low abundance if the antibiotics' effect is strong, but the replicator equation would not reflect this since it describes only the dynamics of relative frequencies. The choice of relative vs absolute densities in empirical data depends on the biological question, e.g. as in the context of microbiome studies [74–76].

Several more complex factors not included in our model are likely to influence the survival of production and resistance mechanisms in the real world. If the antibiotic had a long lifetime in the environment, this would prevent colonisation by sensitive species of a region previously occupied by producer strains. We also assumed that the production and degradation of the chemicals are rapid compared to the cell division timescale. This makes the local antibiotic production proportional to the local concentration of producers in the current time step. Moreover, we assume that the diffusivities of all chemicals of a given type (antibiotic/degrader) are identical. This may change the rules slightly - the effect of more diffusive chemicals is likely to be more important. For example, it may not matter that the interactions concerning a particular antibiotic are destabilising if the antibiotic's diffusivity is such that its effect is not felt strongly.

Additionally, we assume that antibiotics act independently, which does not capture possible synergistic effects commonly seen in microbial communities such as *Streptomyces* [77,78]. These interactions can increase overall toxicity as predicted by independent and additive action and may greatly impact the final ecological outcomes [79,80]. Our framework also restricts the formation of resistance pathways to degradation or intrinsic immunity, while other forms, such as efflux-pump-mediated resistance, can raise extracellular antibiotic levels and intensify selection on sensitive strains [81–83]. Further, production of antibiotics and degrader molecules might themselves be condition-dependent, allowing the microbes to exhibit such costly behaviours only when, e.g. surrounding cell density is high—this is widely known at least for toxin production, see [84,85] for some examples. A natural extension of our work would be to relax the aforementioned

assumptions and consider the proportional cost of producing antibiotics and degradation enzymes with corresponding increasing strength, the effect of limited nutrients, and inter-species conflict and cooperation.

The insights presented in our paper contribute to our knowledge of bacterial coexistence and diversity, with potential implications for drug development. Since our analysis provides rules for stable microbial communities inspired by extant natural consortia, we can apply them to engineer artificial communities. This might have many applications, especially those that benefit from a community perspective, such as bioreactor feeds and microbiome interventions. It is well-known that many bacteria cannot currently be cultured in the laboratory [86]. Understanding more about the community ecology of unculturable bacteria might be central to improving this situation—the rules we prescribe for constructing stable communities might be an answer for unculturable microbes interacting via antibiotics. Natural communities are indeed orders of magnitude larger than our model system [27], however, a detailed understanding of small communities has strong translational relevance and is helpful in understanding their more complicated counterparts. This will require large-scale computational or conceptual models studying eco-evolutionary dynamics in highly diverse communities with higher-order interactions [87,88].

## Materials and methods

All relevant scripts and figure generation pipelines are available at the following public repository: https://doi.org/10.5281/zenodo.17736401

**The mixed-inhibition zone model.** Drawing from the model of Kelsic et al. [49], we begin by considering a microbial community composed of $N$ distinct strains. Each strain can be a producer $P$, degrader $D$, resistant $R$ and sensitive $S$ with respect to $M$ different antibiotics. Let $P, S$ and $D$, and $R$ be $N \times M$ matrices that store the phenotype of each strain concerning each antibiotic: $P_{ij} = 1$ if strain $i$ is a producer of antibiotic $j$, 0 if not. Other matrices are defined similarly. Let $c_p, c_d$, and $c_r$ be the costs of antibiotic production, degradation, and intrinsic resistance. We make the simplifying assumption that the costs for each antibiotic are very similar, or the same (see beginning of Results section). While unrealistic, it allows us to simplify our analysis and isolate the effect of interaction graphs on population dynamics. When considering multiple antibiotics, the base growth rate of a particular strain is reduced in an additive fashion depending on how many antibiotics a strain can produce, degrade, and so forth. If $g$ is the base growth rate of strain $i$, then the effective growth rates $r_i$ are given by,

$$r_i = g - c_p \sum_{j=1}^{M} P_{ij} - c_d \sum_{j=1}^{M} D_{ij} - c_r \sum_{j=1}^{M} R_{ij} \tag{4}$$

Next, we find the probability $p_i^{kill}(t)$ of an individual of strain $i$ dying due to the action of antibiotics at time $t$. This value must be non-zero when strain $i$ is sensitive to antibiotic $j$. We can find the probability $p_i^{kill}(t)$ of dying (due to at least one antibiotic) by using the inclusion-exclusion principle. Let $p_{ij}^{kill}(t)$ be the probability of a strain $i$ individual dying due to the action of antibiotic $j$. Then we have

$$p_i^{kill}(t) = \sum_{j=1}^{M} p_{ij}^{kill}(t) - \sum_{k<l} p_{ik}^{kill}(t) p_{il}^{kill}(t) + \; ... \; + (-1)^{M-1} \prod_{k=1}^{M} p_{ik}^{kill}(t) \tag{5}$$

where the direct summation of the $p_{ij}^{kill}(t)$ is incorrect since these events are not mutually exclusive - an individual of the focal species can simultaneously be inhibited by two or more antibiotics to which it is sensitive. Now we calculate $p_{ij}^{kill}(t)$ for a given strain $i$ and antibiotic $j$. To be killed by antibiotic $j$, an individual of strain $i$ must be sensitive to it, be outside all degrader zones and be inside the production zone of at least one producer. Therefore, we must calculate the probability of being outside and inside these zones. Let $K_P$ and $K_D$ be the effective killing area around each producer and the

effective degradation area around each degrader. We use the Poisson process on the plane as a model for individuals and their (random) position on the Petri plate. If $\lambda$ is the density parameter for this process, the probability of finding zero points in a region of unit area is given by $e^{-\lambda}$.

The abundance $X_i$ can be interpreted as the fraction of the area occupied by strain $i$. But the production (resp. degradation) zones of antibiotic $j$ may arise due to individuals from any strains producing (resp. degrading) this antibiotic, of which there are at most $N$. Consequently, the fraction of area covered by production zones for antibiotic $j$ is $\sum_{k=1}^{N} P_{kj}X_k(t)$ (and similarly $\sum_{k=1}^{N} D_{kj}X_k(t)$ for total degradation zone area). Hence, we have

$$p_{ij}^{kill}(t) = S_{ij}e^{-K_D \sum_{k=1}^{N} D_{kj}X_k(t)}\left(1 - e^{-K_P \sum_{k=1}^{N} P_{kj}X_k(t)}\right) \tag{6}$$

which are then used to find the fitnesses as in the main text 2. One can then search for fixed points of the set of Eqs 1. A fixed point is stable if the maximum absolute value of the eigenvalues of the Jacobian is less than 1.

**Stochastic spatial model.** The lattice is initialised first with each site occupied with independent probability $p$, or left empty. Given a site is occupied, the species identity is chosen uniformly randomly from $1, \ldots, N$. At each time step, we iterate over all sites on the lattice in a random order. If it is occupied, we induce secretion of the relevant antibiotic or degrader chemicals as applicable, depending on the phenotype of this individual. Then, the chemicals diffuse, and we keep track of the cumulative concentrations at each site. Then, we consider the birth and death of this individual. An individual dies with a probability that depends on the antibiotic concentration at the site that it occupies, and the cell is then empty. An individual gives birth only if there is an empty neighbouring site (of which there are 6), and the precise location of the offspring is chosen uniformly from the empty neighbours. This procedure is then iterated for several thousand timesteps. For more details, see Sect A2 in S1 Appendix.

## Supporting information

**S1 Appendix. Removing redundancies in community space, details of the stochastic spatial model, and supplementary figures.**
(PDF)

## Acknowledgments

We gratefully acknowledge Chris Russell for posing a related question on Mathematics Stack Exchange that motivated the formulation of Equation 3, and Mike Earnest for his insightful answer that informed our treatment of this result.

## Author contributions

**Conceptualization:** Gaurav S. Athreya, Chaitanya S. Gokhale, Prateek Verma.

**Data curation:** Gaurav S. Athreya.

**Formal analysis:** Gaurav S. Athreya.

**Funding acquisition:** Gaurav S. Athreya.

**Investigation:** Gaurav S. Athreya.

**Methodology:** Gaurav S. Athreya, Chaitanya S. Gokhale, Prateek Verma.

**Project administration:** Chaitanya S. Gokhale, Prateek Verma.

**Software:** Gaurav S. Athreya.

**Supervision:** Chaitanya S. Gokhale, Prateek Verma.

**Visualization:** Gaurav S. Athreya, Chaitanya S. Gokhale, Prateek Verma.

**Writing – original draft:** Gaurav S. Athreya, Chaitanya S. Gokhale, Prateek Verma.

**Writing – review & editing:** Gaurav S. Athreya, Chaitanya S. Gokhale, Prateek Verma.

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
