## [Decision Letter · Decision Letter 0]

10 Jun 2025

PCOMPBIOL-D-24-02185

Antibiotic-mediated interactions underlying microbial diversity

PLOS Computational Biology

Dear Dr. Verma,

Thank you for submitting your manuscript to PLOS Computational Biology. After careful consideration, we feel that it has merit but does not fully meet PLOS Computational Biology's publication criteria as it currently stands. Therefore, we invite you to submit a revised version of the manuscript that addresses the points raised during the review process.

Please submit your revised manuscript within 60 days Aug 10 2025 11:59PM. If you will need more time than this to complete your revisions, please reply to this message or contact the journal office at ploscompbiol@plos.org. Please include the following items when submitting your revised manuscript:

We look forward to receiving your revised manuscript.

Kind regards,

Samraat Pawar, PhD

Academic Editor

PLOS Computational Biology

Natalia Komarova

Section Editor

PLOS Computational Biology

**Additional Editor Comments :**

We have now received 3 reviews. These are largely positive, but some significant technical revisions are needed. Specifically:

* Reviewer 1 highlights an apparent contradiction: Table 1 lists {PD, SP, DS, DP} as stable whereas Fig 2c shows no 4-strain/2-antibiotic stability; they also ask for quantitative trajectories for all “stable” graphs and a clearer treatment of limit cycles versus fixed points.

* Reviewer 2 requests a formal definition of “cyclicity”, explicit counting of non-redundant graphs, parameter ranges for the 900 (Kp, Kd) sets in Fig 4, and clearer rules governing extensions in Table 1 (why repeated phenotypes are excluded for strains but not for antibiotics).

* Reviewer 3 endorses acceptance as-is.

*
**Essential revisions**
*

* Resolve Fig 2c/Table 1 mismatch and state how redundancy filtering (identical rows/columns) is handled.

* Provide a mathematical definition of cyclicity and prove (or numerically demonstrate) why PSD + cyclicity is sufficient but not PSD alone (e.g. {PP, SS, DD}).

* Add quantitative time series (or phase-plane plots) for every graph deemed stable; note when stability is via limit cycles.

* Document full parameter sweeps (cp, cd, cr, Kp, Kd), initial conditions, and replicate strategy.

* Improve figure/table legends, fix minor label/footnote errors, and harmonise metabolic-cost statements.

*Please also provide point-by-point responses to all the Reviewers' comments.*

**Journal Requirements:**

**Reviewers' comments:**

Reviewer's Responses to Questions

**Comments to the Authors:**

**Please note that two reviews are uploaded as attachments.**

Reviewer #1: Attached in file

Reviewer #2: The work by Athreya et al. investigates the complex interplay of antibiotic production, resistance, and degradation in shaping microbial diversity using a theoretical framework.

The core of the manuscript focuses on identifying specific interaction patterns, or "motifs," that are critical for microbial coexistence in interaction graphs, specifically a "producer-sensitive-degrader" (PSD) motif. The study provides mechanistic insights by establishing exact rules for coexistence in small microbial communities and demonstrates that the PSD motif, particularly when combined with a cyclic interaction structure, is sufficient for stable coexistence in well-mixed populations. The authors also use individual-based simulations to explore the role of this motif in spatially structured populations, finding that community coexistence remains robust across a wide range of antibiotic and degrader diffusivities. The findings emphasize that antagonism, often perceived as a barrier to biodiversity, can drive diversity under specific interaction patterns.

I really like how the manuscript was structured overall - all the sections are drafted well - and enjoyed reading it. I do not have any comments and am happy for this paper to be accepted.

Reviewer #3: The review is uploaded as an attachment.

**Have the authors made all data and (if applicable) computational code underlying the findings in their manuscript fully available?**

Reviewer #1: Yes

Reviewer #2: Yes

Reviewer #3: Yes

PLOS authors have the option to publish the peer review history of their article (what does this mean?). If published, this will include your full peer review and any attached files.

Reviewer #1: No

Reviewer #2: No

Reviewer #3: No

**Figure resubmission:**
---

## [Decision Letter · Decision Letter 1]

18 Nov 2025

Dear Dr Verma,

We are pleased to inform you that your manuscript 'Coexistence rules for small, antagonistically interacting microbial communities' has been provisionally accepted for publication in PLOS Computational Biology.

Best regards,

Natalia L. Komarova

Section Editor

PLOS Computational Biology

Natalia Komarova

Section Editor

PLOS Computational Biology

Reviewer's Responses to Questions

**Comments to the Authors:**

Reviewer #1: I would like to commend the authors for their thorough responses and the revisions that have clarified my previous concerns. I have no major issues remaining and am pleased to recommend this manuscript for publication.

I have only one minor comment regarding the new Figure 5: the upper left and lower left panels appear different, although their reported and  values are similar. This difference is not described in the figure legend, and the authors may wish to verify that the panels and accompanying explanation are consistent.

Reviewer #2: The authors have done a great job of responding to the reviewers' comments.

Reviewer #3: The review is uploaded as an attachment.

**Have the authors made all data and (if applicable) computational code underlying the findings in their manuscript fully available?**

Reviewer #1: Yes

Reviewer #2: Yes

Reviewer #3: Yes

PLOS authors have the option to publish the peer review history of their article (what does this mean?). If published, this will include your full peer review and any attached files.

Reviewer #1: No

Reviewer #2: No

Reviewer #3: **Yes: **Lluís Hernández-Navarro

---

## [Editor Report · Acceptance letter]

PCOMPBIOL-D-24-02185R1

Coexistence rules for small, antagonistically interacting microbial communities

Dear Dr Verma,

I am pleased to inform you that your manuscript has been formally accepted for publication in PLOS Computational Biology. Your manuscript is now with our production department and you will be notified of the publication date in due course.

With kind regards,

Anita Estes
